# CLEARING THE PATH FOR TRULY SEMANTIC REPRESENTATION LEARNING

## ABSTRACT

The performance of $\beta$ Variational Autoencoders ($\beta$-VAEs) and their variants on learning semantically meaningful, disentangled representations is unparalleled. On the other hand, there are theoretical arguments suggesting impossibility of unsupervised disentanglement. In this work, we show that small perturbations of existing datasets hide the convenient correlation structure that is easily exploited by VAE-based architectures. To demonstrate this, we construct modified versions of the standard datasets on which (i) the generative factors are perfectly preserved; (ii) each image undergoes a transformation barely visible to the human eye; (iii) the leading disentanglement architectures fail to produce disentangled representations. We intend for these datasets to play a role in separating correlation-based models from those that discover the true causal structure.

The construction of the modifications is non-trivial and relies on recent progress on mechanistic understanding of $\beta$-VAEs and their connection to PCA, while also providing additional insights that might be of stand-alone interest.

## 1 INTRODUCTION

The task of unsupervised learning of *interpretable* data representations has a long history. From classical approaches using linear algebra e.g. via Principle Component Analysis (PCA) (Pearson, 1901) or statistical methods such as Independent Component Analysis (ICA) (Comon, 1994) all the way to more recent approaches that rely on deep learning architectures.

The cornerstone architecture is the Variational Autoencoder (Kingma & Welling, 2014) (VAE) which clearly demonstrated both high semantic quality as well as good performance in terms of *disentanglement*. Until today, derivates of VAEs (Higgins et al., 2017; Kim & Mnih, 2018a; Chen et al., 2018; Kumar et al., 2017) excel over other architectures in terms of disentanglement metrics. The extent of VAE's success even prompted recent deeper analyses of its inner workings (Rolinek et al., 2019; Burgess et al., 2018; Chen et al., 2018; Mathieu et al., 2018).

If we treat the overloaded term disentanglement to the highest of its aspirations, as the ability to recover the *true generating factors* of data, fundamental problems arise. As explained by Locatello et al. (2019), already the concept of generative factors is compromised from a statistical perspective: two (or in fact infinitely many) sets of generative factors can generate statistically indistinguishable datasets. Yet, the scores on the disentanglement benchmarks are high and continue to rise. This apparent contradiction needs a resolution.

In this work, we claim that all leading disentanglement architectures **can be fooled** by the same trick. Namely, by introducing a **small change of the correlation structure**, which, however, **perfectly preserves the set of generative factors**. To that end, we provide an alternative version of the standard disentanglement datasets in which each image undergoes a modification barely visible to a human eye. We report drastic drops of disentanglement performance on the altered datasets.

On a technical level, we build on the findings by Rolinek et al. (2019) who argued that VAEs recover the *nonlinear principle components* of the datasets; in other words nonlinearly computed scalars that are the sources of variance in the sense of classical PCA. The small modifications of the datasets we propose, aim to change the leading principle component by adding modest variance to an alternative

---

Datasets will be released here: `https://sites.google.com/view/sem-rep-learning`

candidate. The "to-be" leading principle component is specific to each dataset but it is determined in a consistent fashion. With the greedy algorithm for discovering the linear PCA components in mind, we can see that any change in the leading principle component is reflected also in the others. As a result, the overall alignment changes and the generating factors get entangled leading to low disentanglement scores. We demonstrate that, even though the viewpoint of (Rolinek et al., 2019) only has theoretical support for the ($\beta$)-VAE, it empirically transfers to other architectures.

Overall, we want to encourage evaluating new disentanglement approaches on the proposed datasets, in which the generative factors are intact but the correlation structure is less favorable for the principle component discovery ingrained in VAE-style architectures. We hope that providing a more sound experimental setup will clear the path for a new set of disentanglement approaches.

## 2 RELATED WORK

The related work can be categorized in three research questions: i) defining disentanglement and metrics capturing the quality of latent representations; ii) architecture development for unsupervised learning of disentangled representations; and iii) understanding the inner workings of existing architectures, as for example of $\beta$-VAEs. This paper is building upon results from all three lines of work.

**Defining disentanglement.** Defining the term *disentangled representation* is an open question (Higgins et al., 2018). The presence of learned representations in machine learning downstream tasks, such as object recognition, natural language processing and others, created the need to *"disentangle the factors of variation"* (Bengio et al., 2013) very early on. This vague interpretation of disentanglement is inspired by the existence of a low dimensional manifold that captures the variance of higher dimensional data. As such, finding a factorized, statistically independent representation became a core ingredient of disentangled representation learning and dates back to classical ICA models (Comon, 1994; Bell & Sejnowski, 1995).
For some tasks, the desired feature of a disentangled representation is that it is *semantically meaningful*. Prominent examples can be found in computer vision (Shu et al., 2017; Liao et al., 2020) and in research addressing interpretability of machine learning models (Adel et al., 2018; Kim, 2019).
Based on group theory and symmetry transformations, (Higgins et al., 2018) provides the *"first principled definition of a disentangled representation"*. Closely related to this concept is also the field of causality in machine learning (Schölkopf, 2019; Suter et al., 2019), more specifically the search for causal generative models (Besserve et al., 2018; 2020).

**Architecture development.** The leading architectures for disentangled representation learning are based on VAEs (Kingma & Welling, 2014). Despite originally developed as a generative modeling architecture, its variants have proven to excel at representation learning tasks. First of all the $\beta$-VAE (Higgins et al., 2017), which exposes the trade-off between reconstruction and regularization via the additional hyperparameter, performs remarkably well. Other architectures have been proposed that additionally encourage statistical independence in the latent space, e.g. FactorVAE (Kim & Mnih, 2018b) and $\beta$-TC-VAE (Chen et al., 2018). The DIP-VAE (Kumar et al., 2017) suggests using moment-matching to close the distribution gap introduced in the original VAE paper. As the architectures developed, so did the metrics used for measuring the disentanglement quality of representations (Chen et al., 2018; Kim & Mnih, 2018b; Higgins et al., 2017; Kumar et al., 2017).

**Understanding inner workings.** With the rising success and development of VAE based architectures, the question of understanding their inner working principles became dominant in the community. One line of work searches for an answer to the question why these models disentangle (Burgess et al., 2018). Another, closely related line of work shows up and tightens the connection between the vanilla ($\beta$-)VAE objective and (probabilistic) PCA (Tipping & Bishop, 1999) (Rolinek et al., 2019; Lucas et al., 2019). Building on such findings, novel approaches for model selection were proposed (Duan et al., 2020), emphasizing the value of thoroughly understanding the working principles of these methods. On a less technical side, (Locatello et al., 2019) conducted a broad set of experiments, questioning the relevance of models given the variance over restarts and the choice of hyperparameter. They also formalized the necessity of inductive bias as a strict requirement for unsupervised learning of disentangled representations. Our experiments are built on their code-base.

## 3 BACKGROUND

### 3.1 QUANTIFYING DISENTANGLEMENT

Among the different viewpoints on disentanglement, we follow recent literature and focus on the connection between the discovered data representation and a set of *generative factors*. Multiple metrics have been proposed to quantify this connection. Most of them are based on the understanding that, ideally, each generative factor is encoded in precisely one latent variable. This was captured concisely by Chen et al. (2018) who proposed the Mutual Information Gap (MIG), the mean (over the $N_w$ generative factors) normalized difference of the two highest mutual information values between a latent coordinate and the single generating factor:

$$\frac{1}{N_w} \sum_{i=0}^{N_w} \frac{1}{H(w_i)} \left( \max_k I\left(w_i; z_k\right) - \max_{k \neq k'} I\left(w_i; z_k\right) \right),\tag{1}$$

where $k' = \arg\max_\kappa I\left(w_i, z_\kappa\right)$. More details about MIG, its implementations, and an extension to discrete variables can be found in (Chen et al., 2018; Rolinek et al., 2019). While multiple other metrics were proposed such as SAPScore (Kumar et al., 2017), FactorVAEScore (Kim & Mnih, 2018a) and DCI Score (Eastwood & Williams, 2018) (see the supplementary material of Klindt et al. (2020)), in this work, we focus primarily on MIG.

### 3.2 VARIATIONAL AUTOENCODERS AND THE MYSTERY OF A SPECIFIC ALIGNMENT

Variational autoencoders hide many intricacies and attempting to compress their exposition would not do them justice. For this reason, we limit ourselves to what is crucial for understanding this work: the objective functions. For well-presented full descriptions of VAEs, we refer the readers to (Doersch, 2016).

As it is common in generative models, VAEs aim to maximize the log-likelihood objective

$$\sum_{i=1}^{N} \log p(\mathbf{x}^{(i)}),\tag{2}$$

in which $\{\mathbf{x}^{(i)}\}_{i=1}^{N} = \mathcal{X}$ is a dataset consisting of $N$ i.i.d. samples $\mathbf{x}^{(i)}$ of a multivariate random variable $\mathbf{X}$ that follows the true data distribution. The quantity $p(\mathbf{x}^{(i)})$ captures the probability density of generating the training data point $\mathbf{x}^{(i)}$ under the current parameters of the model. This objective is, however, intractable in its general form. For this reason, Kingma & Welling (2014) follow the standard technique of variational inference and introduce a tractable Evidence Lower Bound (ELBO):

$$\mathbb{E}_{q(\mathbf{z}|\mathbf{x}^{(i)})} \log p(\mathbf{x}^{(i)} \mid \mathbf{z}) + D_{KL}(q(\mathbf{z} \mid \mathbf{x}^{(i)}) \parallel p(\mathbf{z})).\tag{3}$$

Here, $\mathbf{z}$ are the latent variables used to generate samples from $\mathbf{X}$ via a parameterized stochastic decoder $q(\mathbf{x}^{(i)} \mid \mathbf{z})$.

The fundamental question of *"How do these objectives promote disentanglement?"* was first asked by Burgess et al. (2018). This is indeed *far from obvious*; in disentanglement the aim to encode a fixed generative factor in *precisely* one latent variable. From a geometric viewpoint this requires the latent representation to be **axis-aligned** (one axis corresponding to one generative factor). This question becomes yet more intriguing after noticing (and formally proving) that both objective functions (2) and (3) are *invariant under rotations* (Burgess et al., 2018; Rolinek et al., 2019). In other words, any rotation of a fixed latent representation results in the same value of the objective function and yet $\beta$-VAEs consistently produce representations that are axis-aligned and in effect are isolating the generative factor into individual latent variables.

### 3.3 RESOLUTION VIA NON-LINEAR CONNECTIONS TO PCA

A mechanistic answer to the question raised in the previous subsection was given by Rolinek et al. (2019). The formal argument showed that under specific conditions which are typical for $\beta$-VAEs (called *polarized regime*), the model locally performs PCA in the sense of aligning the "sources

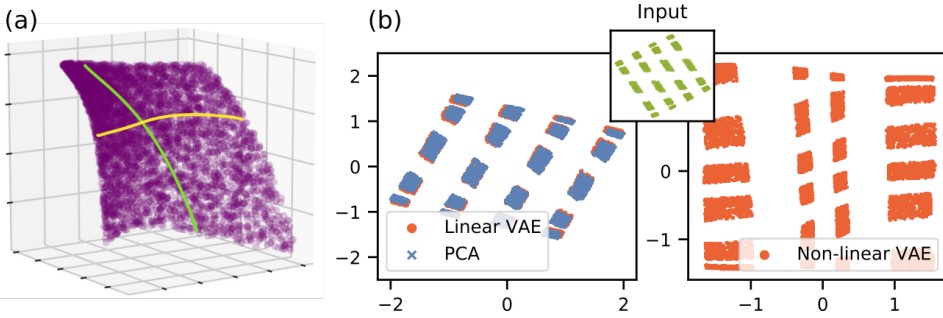

Figure 1: (a) The axes of VAEs latent representation, when decoded back to data space, represent non-linear principal components, as formalized by Rolinek et al. (2019) (figure used with the permission of authors). (b) Distribution of latent encodings for an input distributed as depicted in the inset. The linear VAE's encoding matches the PCA encoding remarkably well (left); both focus on aligning with axes based on explained (global) variance. The non-linear VAE is, however, more sensitive to local variance. It picks up on the natural axis alignment of the microscopic structure. This discrepancy is the driver of the proposed modifications.

of variance" with the local axes. The resulting global alignment often coincides with finding non-linear principal components of the dataset (see Fig. 1). This behavior stems from the convenient but uninformed choice of a *diagonal posterior*, which breaks the symmetry of (2) and (3). This connection with PCA was also reported by Stuehmer et al. (2020), alternatively formalized by Lucas et al. (2019) and converted into performance improvements in an unsupervised setting by Duan et al. (2020).

Since our dataset perturbation method is based on this interpretation of the VAE, we offer further context. In more technical terms, Rolinek et al. (2019) simplify the closed-form KL-term of the objective (3) under the assumption of polarized regime into the form

$$L_{KL}(\mathbf{x}^{(i)}) \approx \frac{1}{2} \sum_{j \in V_a} \mu_j^2(\mathbf{x}^{(i)}) - \log(\sigma_j^2(\mathbf{x}^{(i)})), \tag{4}$$

where $V_a$ is the set of active latent variables, $\mu_j^2(\mathbf{x}^{(i)})$ is the mean embedding of input $\mathbf{x}^{(i)}$ and $\sigma_j^2(\mathbf{x}^{(i)})$ the corresponding term in the diagonal posterior which also plays the role of *noise applied to the latent variable $j$*. This form of the objective, when studied independently, lends itself to an alternative viewpoint. When the only "moving part" in the model is the alignment of the latent representation, i.e. application of a rotation matrix, the induced optimization problem has the following interpretation[1]:

> *Distribute a fixed amount of noise among the latent variables such that the $L^2$ reconstruction error increases the least.*

This extracted problem can be solved in closed form and the solution is based on *isolating sources of variance*. Intuitively, **important scalar factors** whose preservation is crucial for reconstruction, **need to be captured with high precision** (low noise). For that, it is economic to isolate them from other factors. This high-level connection with PCA is then further formalized by Rolinek et al. (2019).

One less obvious observation is that the "isolation" of different sources of variance relies on the accuracy of the linearization of the decoder at around a fixed $\mu(\mathbf{x}^{(i)})$. Since in many datasets the local and global correlation structure is nearly identical, $\beta$-VAE recover sound global principle components. If, however, the local structure obeys a different "natural" alignment, the Variational autoencoder prefers it to the global one as captured in a synthetic experiment displayed in Fig. 1.

The sensitivity of $\beta$-VAE's global representation to small local changes is precisely our point of attack.

---

[1]the precise formal statement can be found in (Rolinek et al., 2019, Equations (22) and (23))

## 4 METHODS

The standard datasets for evaluating disentanglement all have an explicit generation procedure. Each data point $\mathbf{x}^{(i)} \in \mathcal{X}$ is an outcome of a generative process $g$ applied to input $\mathbf{w}^{(i)} \in \mathcal{W}$. Imagine that $g$ is a function rendering a simple scene from its specification $w$ containing *as its coordinates* the background color, foreground color, object shape, object size etc. By design, the individual generative factors are statistically independent in $\mathcal{W}$. All in all, the dataset $\mathcal{X} = \big(\mathbf{x}^{(1)}, \mathbf{x}^{(2)}, \ldots, \mathbf{x}^{(n)}\big)$ is constructed with $\mathbf{x}^{(i)} = g(\mathbf{w}^{(i)})$, where $g$ is a mapping from the generative factors to the corresponding data points.

In this section, we introduce a modification $\widetilde{g}$ of the generative procedure $g$ that barely distorts the resulting data points. In particular, for each $\mathbf{x}^{(i)} \in \mathcal{X}$, we have

$$d\big(\mathbf{x}^{(i)}, \widetilde{g}(\mathbf{w}^{(i)})\big) \le \varepsilon \tag{5}$$

under some distance $d(\cdot, \cdot)$.

How to design $\widetilde{g}$ such that despite an $\varepsilon$-small modification VAE-based architectures will create an entangled representation? Following the intuition from Sec. 3.3 and Fig. 1, we "misalign" the local variance with repect to the global variance in order to promote an alternative (entangled) latent code.

To avoid hand-crafting this process we can exploit the following observation. VAE based architectures suffer from large performance variance over e.g. random initializations. This hints to the existing ambiguity: two or more candidates for the latent coordinate system are competing minima of the optimization problem. Some of these solution are "bad" in terms of disentanglement. Below we elaborate on how to foster these bad solutions.

It should be noted that our dataset modifications are not an implementation of (Locatello et al., 2019, Theorem 1). We do not modify the set of generative factors. Rather, we slightly perturb the generation process in order to target a specific subtlety in the inner working of VAEs.

Overall, our modification process has three steps:

(i) Find the most entangled alignment that a $\beta$-VAE produces over multiple restarts and retrieve its first principle component, denoted by $s$.

(ii) Fix a "noise pattern" that is highly tied with $s$.

(iii) Add the noise pattern with suitable magnitude $\varepsilon$ to each image.

### 4.1 CHOICE OF FOSTERED LOCAL PRINCIPAL COMPONENT

Over multiple restarts of $\beta$-VAE we pick one with the lowest MIG score. This gives us an entangled alignment that is expressible by the architecture. It's first principle component is captured by $s \colon \mathcal{X} \to \mathbb{R}$ computed as the value of the latent coordinate $j$ that has the least noise $\sigma_j^2$ injected (averaged over the dataset):

$$s\big(\mathbf{x}^{(i)}\big) = \mathrm{enc}\big(\mathbf{x}^{(i)}\big)_j \quad j = \arg\min_k \big\langle \sigma_k^2 \big\rangle. \tag{6}$$

This procedure of retrieving the most "important" latent coordinate is consistent with (Higgins et al., 2017) and (Rolinek et al., 2019). The analogy to PCA is that the mapping $s(\mathbf{x}^{(i)})$ gives the first coordinate of $\mathbf{x}^{(i)}$ in the new (non-linear) coordinate system.

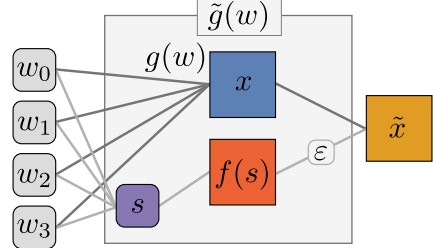

Figure 2: A schematic visualization of the image generation process. The set of ground truth generating factors stays untouched by the modification.

### 4.2 MANIPULATION PATTERNS

We will now describe the modification procedure assuming the data points are $r \times r$ images. The manipulated data-point $\widetilde{\mathbf{x}}^{(i)}$ is of the form $\widetilde{\mathbf{x}}^{(i)} = \mathbf{x}^{(i)} + \varepsilon f\big(s(\mathbf{x}^{(i)})\big)$ where the mapping $f \colon \mathbb{R} \to \mathbb{R}^r \times \mathbb{R}^r$ is constrained by $\|f(s)\|_\infty \le 1$ for every $s$. Then inequality (5) is naturally satisfied for the maximum norm.

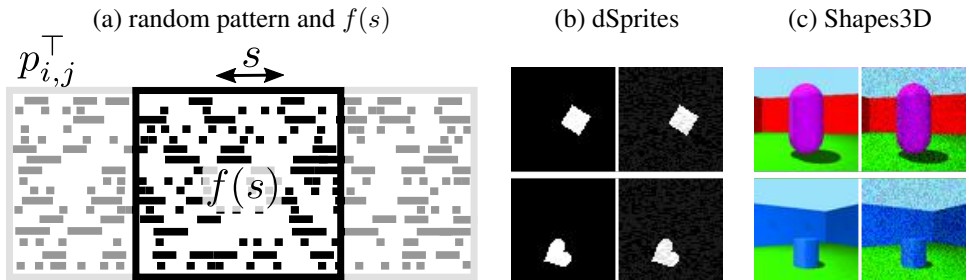

Figure 3: Dataset modification. (a) Construction of the manipulation pattern (transposed visualization). (b,c) Visual comparison between original and perturbed samples side-by-side.

At the same time, we aim to produce large local changes. More technically, the *local expansiveness factor*

$$\alpha_\delta(s) = \inf_{|\delta'| < \delta} \frac{\|f(s) - f(s + \delta')\|}{|\delta'|} \tag{7}$$

should be as high as possible everywhere. Additionally, the mapping should be injective so that the resulting modifications contains information about $s$.

For the sake of simplicity, we accomplish the above requirements (with high probability) by vertically shifting a fixed random pattern. Specifically, for fixed randomly sampled $p_{i,j} \in \{-1, 1\}$ for $i, j \in \mathbb{N}$ and "column factors" $c_j \in \{1, 2, 3, 4\}$, we set $f$ as

$$(f(s))_{i,j} = p_{i',j}, \qquad \text{where} \qquad i' = \text{round}(s/c_j) + i. \tag{8}$$

This random pattern is moving along the first coordinate with $s$ while creating consecutive blocks of length $c_j$ in column $j$ in order to incorporate information about $s$ on multiple scales.

## 5 EXPERIMENTS

In order to experimentally validate the soundness of the manipulations, we need to demonstrate the following:

1. **Effectiveness of manipulations.** Disentanglement metrics should drop on the modified datasets across architectures and datasets.
2. **Qualitative analysis.** The changes in local and global statistics of the dataset should match the intuition fleshed out in Sec. 3.3.
3. **Comparison to a trivial modification.** Instead of the proposed method, we modify with uniform noise of the same magnitude. The disentanglement scores for the algorithms on the resulting datasets should not drop significantly.
4. **Robustness.** The new datasets should be hard to disentangle even after retuning hyperparameters of the original architectures.

### 5.1 EFFECTIVENESS OF MANIPULATIONS

We use the scalar $s = s(\mathbf{x}^{(i)})$ as described in Sec. 4.1 and embed it in the data-space via the manipulation $f(s)$ described in Sec. 4.2. We deploy this approach on two datasets: Shapes3D (Burgess & Kim, 2018) and dSprites (Matthey et al., 2017), leading to manipulations as depicted in Fig. 3(b,c).

Four VAE based architectures and a regular autoencoder are evaluated on both the original and manipulated dataset using literature regularization strength (or better). Other hyperparameters are taken from the disentanglement library (Locatello et al., 2019). For the sake of simplicity and clarity, we restricted the latent space dimension to be equal the number of ground truth generative factors.

The resulting MIG scores are listed in Tab. 1. Over all models and datasets, the disentanglement quality is significantly reduced.

Table 1: MIG Scores for the unmodified and the modified dataset. Each setting was run with 10 distinct random seeds.

|  |  | **AE** | $\beta$-**VAE** | **TC-$\beta$-VAE** | **Factor-VAE** |
|---|---|---|---|---|---|
| **dSprites** | **orig.** | $0.06 \pm 0.03$ | $0.23 \pm 0.08$ | $0.25 \pm 0.08$ | $0.27 \pm 0.11$ |
|  | **mod.** | $0.05 \pm 0.03$ | $0.07 \pm 0.04$ | $0.14 \pm 0.05$ | $0.11 \pm 0.05$ |
| **Shapes3D** | **orig.** | $0.09 \pm 0.06$ | $0.60 \pm 0.31$ | $0.58 \pm 0.20$ | $0.27 \pm 0.18$ |
|  | **mod.** | $0.05 \pm 0.05$ | $0.02 \pm 0.01$ | $0.28 \pm 0.23$ | $0.11 \pm 0.09$ |

## 5.2 QUALITATIVE ANALYSIS

The proposed manipulations are by definition of constraint (5) small in terms of overall variance, but chosen to have a large local expansiveness factor. We want quantify both and the order in which the $\beta$-VAE encodes the generating factors. In addition to the existing generating factors and their influence on the dataset, we also want to evaluate the manipulation applied in the previous experiment. By adding an additional generating factor $s$, a uniform random variable with zero mean and unit variance, we isolate the effect of the manipulation from the other factors.

We approximate the *explained global variance* of a generating factor $\mathbf{w}^{(i)}$ by extensively sampling

$$\text{var}_{w_j} \mathbb{E}_{\mathbf{x}^{(i)}(\mathbf{w}^{(i)}),\, \mathbf{w}_j^{(i)}=w_j} \left( \mathbf{x}^{(i)} \right). \tag{9}$$

Due to the discrete nature of the generating factors, the *local expansiveness factor* is calculated for $\delta'$ equal the step size of the position generating factors. We train a $\beta$-VAE on the same modified dataset and report the *"information level"* (which determines the order of the principle components) of the latent coordinate that encodes each individual generating factor, averaged over 10 restarts. The reported quantity is $1 - \mathbb{E}\left(\sigma_i^2\right)$, i.e. the lack of noise in the latent coordinate.

Figure 4 (a) and (b) show the local expansiveness factor and the explained global variance for different values of $\varepsilon$. The local expansiveness of $s$ exceeds all other generating factors at $\varepsilon \approx 0.1$, long before it's explained global variance becomes dominant. At the same point $s$ becomes the dominant coordinate in the latent encoding, as shown in Fig. 4 (c).

In conclusion, $\beta$-VAE chooses to encode $s$ as the first principle component for $\varepsilon \approx 0.1$ which is exactly when $s$ starts to dominate local statistics but long before it dominates global statistics. This is consistent with our prediction in Sec. 3.3.

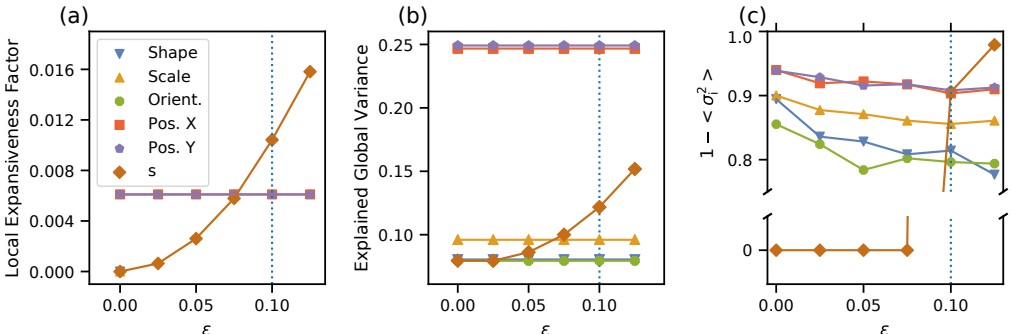

Figure 4: A comparison of the local expansiveness factor, the explained global variance and the $\beta$-VAE noise distribution. (a) Sampled local variance for different values of $\varepsilon$. Restricted to the position factors (curves are overlapping) since the sampling in the generating factor space is discrete and varying step sizes are inexpressive. (b) Variance for all generating factors and $s$. (c) The mean latent noise standard deviation is an indicator for the relevance for reconstruction. Increasing $\varepsilon$ makes $s$ the prime encoded coordinate. For $\varepsilon \leq 0.075$, $s$ was not encoded due to it's noisy nature.

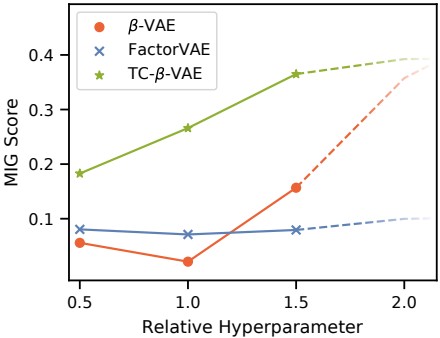 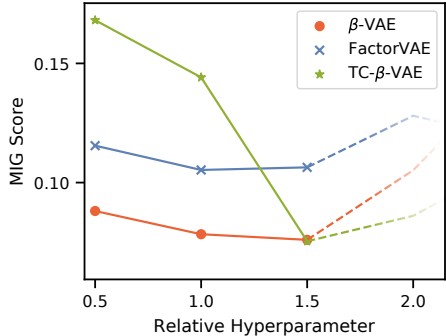

Figure 5: MIG scores for scaled literature hyperparameters over 10 restarts. The dashed lines show the average number of active units. Less than 5 (left, dSprites) or 6 (right, shapes3D) active units mean the architectures overpruned and can not recover the true generative factors.

## 5.3 NOISY DATASETS

We replace our modification by contaminating each image with pixel-wise uniform noise between $[-\varepsilon, \varepsilon]$. The value of $\varepsilon$ is fixed to the level of the presented manipulations. Table 2 provides the results for the same five architectures as used before. The lack of structure in the contamination does not affect the performance in a guided way and leads to very little effect on 3DShapes. The impact on dSprites is, however, noticeable. Due to the comparatively small variance among dSprites images, the noise shifts the balance between the reconstruction loss and the regularizers. We performed a grid search over $\beta$ and recovered the performance on the noisy dataset ($\beta^\star = 16$), the same can be expected for the other architectures.

Table 2: MIG scores on original and a noisy version of the datasets with literature hyperparameters. $\beta^\star$ was tuned to the noisy dataset. Noise cannot explain the full loss in disentanglement.

|  |  | $\beta$-**VAE** | $\beta^\star$-**VAE** | **TC-$\beta$-VAE** | **Factor-VAE** |
|---|---|---|---|---|---|
| **dSprites** | **orig.** | $0.23 \pm 0.08$ | | $0.25 \pm 0.08$ | $0.27 \pm 0.11$ |
| | **noisy** | $0.09 \pm 0.05$ | $0.23 \pm 0.13$ | $0.20 \pm 0.04$ | $0.16 \pm 0.08$ |
| **Shapes3D** | **orig.** | $0.60 \pm 0.31$ | | $0.58 \pm 0.20$ | $0.27 \pm 0.18$ |
| | **noisy** | $0.66 \pm 0.05$ | | $0.60 \pm 0.11$ | $0.33 \pm 0.20$ |

## 5.4 GRIDSEARCH ON HYPERPARAMETERS

We run a grid searches over the hyperparameter for each architecture. The results are illustrated in Fig. 5. Overall our modifications seem mostly robust for adjusted hyperparameters. On both datasets, significant increase in the regularization strength allows for some recovery. More thorough analysis reveals that this effect starts only once the models reach a level of over-pruning, which is a behavior well known to practitioners. The dashed curves mark the region in which the active latent space dimension (number of coordinates for which $\mathbb{E}\left(\sigma_i^2\right) < 0.8$) shrunk significantly. This effect goes along with decreased reconstruction quality and also intrinsically prevents the models from recovering all true generative factors and as such renders this area uninteresting.

## 6 CONCLUSION

We have shown that the success of $\beta$-VAE based architectures is solely based on the structured nature of the datasets they are being evaluated on. The goal of truly semantic representation learning requires an alternative benchmark to excel at, free of similar correlation-based artifacts. We propose to the community a set of novel datasets and hope that future architectures become capable of learning truly semantic representation.

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
