# OpenReview forum: "Clearing the Path for Truly Semantic Representation Learning"
_ICLR.cc/2021/Conference — Reject_

### Official Review · AnonReviewer3 · 2020-10-23
**Interesting work**

**Rating:** 5
**Confidence:** 2

**Review:**


This work aims to demonstrate that VAE-based architectures can take advantage of inherent correlation between data to produce representations. They claim that small perturbations of the data prevent these architectures from taking advantage of such correlations resulting in failure to produce disentangled representations. This work therefore claims that the perturbed datasets should be used to design and learn models that can discover the true structure of the data

This work highlights a characteristic of leading models for disentanglement. It demonstrates that these models are taking advantage of the correlation in the data and not fully learning the true structure. Therefore, while the current models are certainly useful, models that can learn the true structure might be even more beneficial.

They take a structured approach by perturbing the data in a manner such that the local variance is `misaligned' from the global variance. They demonstrate empirically that the performance of the models goes down for the data set when it is perturbed in the proposed structured manner. This also shows the merit of the perturbation procedure as adding uniform noise to pixel doe not produce the similar drop in performance.


The work, therefore, does seem to have merit. However, there are some clarifications that are needed.


Figure 1 is a bit unclear. If 1a and 1b is supposed to show a distribution then why does the vertical axis have -ve values? I am basically not fully sure what exactly is being depicted by 1a and 1b. They look similar to the raw data in the inset. If its just the principal components then should it not be directions and not a patch? Perhaps the figure can be clarified.

Eq8 is a bit unclear. How is $c_j$ selected from the set $\{1,2,3,4\}$. Also assuming s = s(x^{(i)}), is it possible that $i'$ = $i$ for all $i$ ?. Furthermore I am a bit confused about the notations. For example, $f:\mathbb{R} \rightarrow \mathbb{R}^{r} \times \mathbb{R}^{r}$. This seems to imply that $f(s) = (v,u)$ where $v,u \in  \mathbb{R}^{r}$. In eq8, however it seems that f(s) is a matrix of scalars? This would imply that $f:\mathbb{R} \rightarrow P^{2}$ where $P = \{ -1,1 \}$. In addition to this, why do i,j belong to \mathbb{N} (and not related to the size of input x).

In the context of Figure 4, How are the $1 - \mathbb{E}(\sigma_{i}^{2})$ calculated for shape scale orient, PosX and PosY. Are they calculated using the reconstruction?


Were any other types of f() explored. What influence does the choice of f() (that abides by all the conditions in Section 4.2) have on the performance. If it is specifically the proposed Eq8 that produces this influence then what might be the major characteristics of the f() that produce the drop in performance?

---

### Official Review · AnonReviewer2 · 2020-10-28
**Not an indictment of disentanglement datasets or models**

**Rating:** 5
**Confidence:** 4

**Review:**

This paper describes a corruption–of image data in commonly used disentanglement benchmarks– which hurts the MIG disentanglement score of top models. The corruption additively encodes an *entangling* factor into each image. The entangling factor is obtained from a poorly disentangled beta-VAE model trained on the original data.

My chief criticism of the paper is I’m not convinced it discredits existing disentanglement datasets or models. (Nor does the paper clear the path for truly semantic representation learning, but I will reserve my criticism of the paper title.) Perhaps it discredits the MIG metric, but that does not seem sufficient to me.

The authors show that models like beta-VAE, TC-beta-VAE, and Factor-VAE are compelled to encode s because it is crucial to reduce the reconstruction error. But s happens to be correlated (and is a deterministic function of) the other generative factors w_j. Therefore it seems redundant to encode. Nevertheless, it is possible to make a case that the models are doing their job correctly (if they do encode the other generative factors as independent latents, which is not clear from the results in the paper). Consider Suter et al. (2019)’s proposal suggesting disentangled representations should reflect causally independent generative mechanisms (influences) on the data. The entangling factor s certainly has an independent generative effect on the images. So perhaps the authors mean to indict mainstream definitions/expectations of disentanglement instead?

Here are some questions and requests for additional details:
1) How does your s relate to the generative factors (on each dataset)? Could you provide its mutual information with position, scale, and the other factors?
2) Could you offer a breakdown (across generative factors) of the MIG scores on the corrupted data?
3) Is there a single latent that encodes for s typically? What if you exclude this latent when computing the corrupted MIG scores?
4) What would be other reasonable choices of s (other than the most important latent of a poorly disentangled beta-VAE model) which would break disentangling? What choices of s would not break disentangling?

Minor suggestions:
- The corruption itself could be better motivated and explained. The Related Work and Background sections, though extensive, don’t immediately suggest your corruption function.
- It would be worth explaining the “local expansiveness factor” and “explained global variance” before the main analysis. Equation 9 is also hard to grasp. Could you please unpack it?
- “Lack of noise” can be formalized as precision or 1/sigma**2.

Strengths:
- The analysis around Figure 4 (mainly that epsilon around 0.1 is where s starts to be encoded, despite its relatively low effect on the global variance).
- Comparison of several models showing a consistent effect of the suggested corruption.
- Results from varying the regularization hyperparameters.

References:
Suter, R., Miladinovic, D., Schölkopf, B., & Bauer, S. (2019, May). Robustly disentangled causal mechanisms: Validating deep representations for interventional robustness. In International Conference on Machine Learning (pp. 6056-6065). PMLR.

---

### Official Review · AnonReviewer4 · 2020-10-28
**Margianl contributions and unconvincing experiments.**

**Rating:** 3
**Confidence:** 2

**Review:**

This paper shows a new method to hurt the disentanglement of Variational AutoEncoders (VAEs) by only slightly modifying the images. It conducts experiments and shows that the disentanglement metrics drop significantly by this carefully designed perturbation compared to uniformly noise, and the results are consistent across different variants of VAEs and various datasets. These results show that the success of VAEs are based on the structured nature of the datasets, and the community needs to find new datasets to learn the semantic representations.

The idea proposed by this paper is interesting. It believes that the disentanglement of beta-VAEs comes from the local PCAs, which coincides with global coordinate alignments, and this strange behavior stems from the assumption of diagonal posterior. Thus the global representation of beta-VAEs is sensitive to local changes. Also, this paper comes with a detailed background description, which is very much self-contained. The analysis of the results is also exhaustive, and it well supports the claim of this paper.

However, the methodology contribution of this paper is marginal and the method it adopts seems over-simplified. The method is not technical and is nothing but noise to the generation process, and the design of the noise is not elaborate, either. E.g., instead of a detailed noise on all coordinates, the paper only applies this on the "most important latent coordinate", which is less convincing and lacks an explanation.

Also, the conclusions of this paper are already well-discussed by previous works, and the experiments of this paper seem like a verification. The section 3 of this paper revisits many previous results but no new idea is proposed across this paper.

More metrics should be used. As described in the paper, various metrics have been proposed: SAPScore, FactorVAEScore, DCI Score, but this paper only adopts the Mutual Information Gap, which is neither typical nor representative. Also, the correlation between these metrics is not very much high according to fig2 of Locatello et al. Given all these metrics are already implemented by disentanglement_lib and the codebase of this paper is based on that paper, it's not too hard to show all of them.

Some connections between section 3 and the remainder of the paper are not clear. E.g. this paper argues the importance of local and global alignments in section 3, but it's unclear which part of section 4 corresponds to this point. The term "local expansiveness" is used in section 4 but was not discussed in section 3. Some complex ideas were discussed in section 3 but never touched again.

It is said that the modification to the images is barely visible to human eyes, but clearly, the images of fig 3 are heavily blurred and it's easy to tell the difference. This perturbation is not "a small modification" and is human sensible in contrast to other attack/defense papers. The authors should show out a measure of perturbations, e.g. the relative magnitude of epsilon.

No baseline works are introduced. The authors only compare their results with a model that works with a random baseline, and it's not convincing.

Questions:

At the end of section 4, you propose a simplification of the method by a shifting window. Why don't you just apply the modification for the whole image? Is there a specific reason that prevents you from doing so?

Presentation suggestions:

The paper spends almost 3 pages on background research, but only 1 page on its method. It'd better put more stuff in section 4 and move some content from section 3 to section 2 or do a better summarization.

Typo:

The third line of section 4.1: it's -> its

---

### Official Review · AnonReviewer1 · 2020-10-29
**Benefit of the proposed data modification is not clear**

**Rating:** 4
**Confidence:** 5

**Review:**

The authors first describe the shortcomings of existing work for disentanged representation learning and the general problem of unidentifiability. Here however, the presentation could be more precise. For example, they state that the log-likelihood objective (2) and the evidence lower bound (3) would be "invariant under rotations", and give (Burgess et al., 2018; Rolinek et al., 2019) as a reference. However, the log likelihood or the evidence lower bound themselves are not rotationally invariant per-se. Instead these objectives are invariant under rotations for the standard VAE with a rotationally invariant prior p(z) (e.g. the standard normal distribution).

The authors then discuss a possible shortcoming of VAE based approaches: their sensitivity to local covariance statistics in the dataset. From this perspective the authors motivate their idea to modify existing datasets by adding a noise pattern to the dataset that is highly correlated with the first principle component identified by the model when trained on the standard dataset.

Essentially the authors crafted a specialised attack against VAE based approaches for "disentangled" representation learning by adding another source that is highly correlated with the main source of variation of the data. Any approach similar to PCA (perhaps even to ICA), would fail on this task of separating these two highly correlated signals.

Thus I am not convinced about any benefits of the proposed data modification and how this work would "clear the path for truly semantic representation learning". The authors do not demonstrate how a method that could decorrelate the added noise signal from the original dataset would have any further benefits.


Positive Points:

+ Criticism of current state-of-the-art methods for learning disentangled representations is valid

Negative Points:

- Mathematical background needs to be more precise

- Does not take into account recent advances in self supervised representation learning, e.g. noise-contrastive learning and Nonlinear ICA [1]

- Goal of the paper is not clear

- Benefit of the proposed data modification is not clear

[1] Ilyes Khemakhem, Diederik P. Kingma, Ricardo P. Monti, and Aapo Hyvärinen. Variational Autoencoders and Nonlinear ICA: A Unifying Framework. AISTATS2020

---

### Author Response · Authors · 2020-11-23
**Thanks to the reviewers**

We thank the reviewers for their effort. Given the broad range of feedback and the substantial nature of the raised concerns, we decided to not proceed with this submission.

---

### Decision · Program_Chairs · 2021-01-07
**Final Decision**

**Decision:**

Reject

**Comment:**

Since the authors have decided to withdraw this submission, it has been rejected from the conference.